# The association between parent-child communication and adolescents' non-cognitive abilities: An examination of the chain mediating effect

Minghan Cai[1], Huijie Guo[2]*, Dailong Fang[3], Yao Zheng[4], Weifeng Guo[2], Zhengmei Lin[5], Zhiqiang Zheng[1]*

1 College of Physical Education, Jimei University, Xiamen,China, 2 School of Physical Education and Sport Science, Fujian Normal University, Fuzhou, China, 3 Police Physical Education and Training Department, Fujian Police College, Fuzhou, China, 4 Fuzhou Sixteenth Middle School, Fuzhou, China, 5 School of Physical Education, Fujian Polytechnic Normal University, Fuqing, China

* guohuijie@fjnu.edu.cn (HG); 202061000042@jmu.edu.cn (ZZ)

## Abstract

This study aims to explore the interplay between parent-child communication, social support, physical exercise and non-cognitive abilities among adolescents. Using data from the China Education Panel Survey (2014–2015) with a nationally representative sample of 5,055 eighth-grade students. Analyses were conducted via Stata 17.0 and SPSS 26, employing chain mediation tests and bootstrap analysis. The results were as follows:1) parent-child communication directly contributes to the enhancement of adolescents' non-cognitive abilities, facilitates their access to social support, and promotes their participation in physical exercise; 2) parent-child communication indirectly improves adolescents' non-cognitive abilities through the chain mediating effect of social support and physical exercise; 3) the positive effect of parent-child communication on non-cognitive abilities is stronger among urban adolescents compared to rural adolescents, and more pronounced among only children than non-only children. In conclusion, parent-child communication not only directly and positively predicts adolescents' non-cognitive abilities but also exerts indirect effects via two single mediators (social support and physical exercise) and a chain mediation mechanism combining both mediators.

## Introduction

In recent years, with the emergence of new human capital theory, non-cognitive abilities have become a focal point in academic and policy discussions because of their profound impact on adolescent development [1,2]. Non-cognitive abilities, often encompassing traits such as emotional regulation, resilience, and interpersonal skills, are critical for adolescents' success in both academic and non-academic domains

**Data availability statement:** The data underlying the results presented in the study are available from https://doi.org/10.3886/E239282V1.

**Funding:** This work was supported by the National Office for Philosophy and Social Science of China (Grant No. 22BTY114 awarded to Huijie Guo; Grant No. 23BTY021 awarded to Zhiqiang Zheng).

**Competing interests:** The authors have declared that no competing interests exist.

[3]. As a core competency of the 21st century, non-cognitive abilities are regarded as implicit human capital that not only influences adolescents' academic achievements but also shapes their career opportunities and social mobility in the labor market [4]. The World Development Report 2018 by the World Bank further highlights their importance, identifying them as critical skills for students' development [5]. Adolescence is a critical period for the formation of non-cognitive abilities, as it serves as a pivotal stage for cultivating the character traits and competencies necessary for lifelong development [6]. Research consistently shows that investing in these abilities is among the most effective strategies for supporting youth development, linking them to intellectual advancement and long-term academic success [7,8]. Within the family context, parents play a vital role, with active parental involvement significantly enhancing adolescents' non-cognitive abilities [9–11]. Among various parental behaviors, parent-child communication has emerged a particularly influential factor that strongly shapes the development of non-cognitive abilities [12]. Exploring the relationship between parent-child communication and adolescents' non-cognitive abilities is therefore highly important. Moreover, studies suggest that adolescents' non-cognitive abilities are influenced by broader social and individual factors, such as social support [13], physical exercise [14], and school climate [15]. To fully understand how parent-child communication affects non-cognitive abilities, it is essential to account for these additional factors.

This study utilizes data from the 2014–2015 China Education Panel Survey (CEPS) to investigate the multifaceted relationship between parent-child communication and adolescents' non-cognitive abilities. This study explores how various aspects of parent-child interaction contribute to the development of these abilities through complex and interrelated pathways. By adopting this comprehensive approach, this research aims to provide deeper insights into the complex determinants of non-cognitive abilities. Additionally, it underscores the critical role of parents in fostering adolescents' holistic development and emphasizes the value of cultivating close and harmonious parent-child relationships.

## Literature review and research hypotheses

### Linking parent-child communication to non-cognitive abilities

Parent-child communication refers to the daily interactions between parents and children and is a key form of intrafamily parental involvement [16,17]. This type of communication plays a significant role in the development of adolescents' non-cognitive abilities, particularly in providing emotional support and promoting psychological well-being. From the perspective of human capital development, parent-child communication is regarded as an important indicator of parental companionship, with a substantial influence on the formation of both cognitive and non-cognitive abilities. Empirical evidence demonstrates that parent-child communication contributes positively to children's holistic development, particularly in fostering non-cognitive abilities. Specifically, mother-child communication exhibits significant positive effects on adolescents' extraversion ($r = 0.022$, $p < 0.01$) and openness ($r = 0.038$, $p < 0.01$) [18]. According to self-determination theory, individual growth requires the fulfillment

of three basic psychological needs: autonomy, competence, and relatedness. Parent-child communication enables parents to provide emotional support, satisfying adolescents' need for relatedness, thereby enhancing their self-efficacy and psychological adaptability [19]. Shared family meals, as an essential form of parent-child communication, have been shown to positively impact adolescents' physical health, mental well-being, cognitive abilities, and academic performance. Multiple studies have highlighted a strong positive correlation between the frequency of family meals and adolescents' psychological health. Collectively, these findings support the positive influence of parent-child communication on adolescents' non-cognitive abilities [20,21]. These studies collectively underscore the positive relationship and impact between parent-child communication and adolescents' non-cognitive abilities. Accordingly, this study proposes Hypothesis 1: Parent-child communication is significantly associated with adolescents' non-cognitive abilities.

## The mediating role of social support

Social support is defined as the actual or perceived instrumental and expressive support provided by communities, social networks, and close relationships [22]. Poor parent-child communication can weaken adolescents' perception of family support, thereby hindering the construction of their social support systems [23]. Prior studies consistently demonstrate that high-quality parent-child communication enhances adolescents' access to social support, thereby protecting mental health and strengthening non-cognitive abilities [24,25].

According to social cognitive theory, external support influences the development of non-cognitive abilities by shaping individual self-efficacy [26]. Adolescents primarily receive social support from family (parents) and school (teachers and peers) contexts. Beyond the family, teacher emotional support has been linked to higher levels of agreeableness and extraversion [27], while peer support can buffer disadvantages faced by vulnerable groups such as left-behind children [28]. Taken together, these findings highlight that social support is an important pathway through which interpersonal interactions foster adolescents' non-cognitive growth. However, few studies have focused on the mediating role of social support in the relationship between parent-child communication and adolescents' non-cognitive abilities. Given the facilitative effect of parent-child communication on social support and the positive impact of social support on adolescents' non-cognitive abilities, this study proposes Hypothesis 2: Social support mediates the relationship between parent-child communication and adolescents' non-cognitive abilities.

## The mediating role of physical exercise

Physical exercise refers to activities aimed at improving physical and mental health through leisure, recreational, or and fitness-oriented pursuits. Parental involvement creates a conducive family educational environment that positively influences children's participation in physical exercise. According to intergenerational transmission mechanisms and vertical model theories, parent-child communication promotes adolescent physical exercise through dual channels [29]. The first is cognitive transmission, whereby parents convey behavioral preferences and value orientations through direct education and interaction, internalizing exercise values into adolescents' behavioral norms [30]. Empirical studies confirm that parent-child communication increases adolescents' exercise time and promotes physical activity by enhancing health awareness [31]. The second channel is behavioral synchronization, in which parents' regular exercise behaviors create a supportive family atmosphere and foster exercise habits in children through observational learning [32]. Evidence shows that compared with non-exercising parents, those exercising 1–2 times or ≥3 times per week substantially increase the likelihood of their children engaging in moderate-to-vigorous physical activity (MVPA), with stronger effects observed for higher parental exercise frequency [33].

From a bio-psycho-social model perspective, physical exercise promotes non-cognitive ability development through multiple mechanisms. Biologically, it regulates neuroendocrine activity and enhances emotional stability. Psychologically, it cultivates perseverance, communication, and adaptability [34], while socially, it strengthens trust, prosocial behavior, and engagement in collective activities [35]. Drawing on ecological systems theory, physical exercise also represents a socially

embedded context shaped by family, school, and community interactions, within which adolescents develop confidence, self-regulation, and resilience [36]. Thus, it is proposed Hypothesis 3: Physical exercise mediates the relationship between parent-child communication and adolescents' non-cognitive abilities.

### The role of social support and physical exercise

Parent-child communication, social support, and physical exercise all have an impact on adolescents' non-cognitive abilities. Existing research indicates that social support significantly enhances adolescents' participation in physical exercise [37]. Specifically, support from parents is beneficial for adolescents to improve their Body Mass Index (BMI) and foster a positive attitude towards engaging in physical exercise [38]. Attachment theory suggests that effective parent-child communication provides adolescents with emotional support and guidance [39]. This not only directly influences the breadth and quality of their social support networks but also indirectly affects their motivation for physical exercise. Parental communication and care strengthen adolescents' ability to interact with peers, allowing them to gain greater support from peer relationships, which in turn fosters stronger beliefs and enthusiasm for physical exercise [40]. From the perspective of Self-Determination Theory (SDT), social support plays a crucial role in satisfying adolescents' basic psychological needs for relatedness, competence, and autonomy [41]. When these psychological needs are adequately satisfied, adolescents are more likely to internalize favorable attitudes toward physical activity, which in turn strengthens their intrinsic motivation to engage in and sustain such behaviors [42]. Previous research has demonstrated that parental social support can effectively foster children's autonomous motivation and decision-making capacity for engaging in physical activities [43]. Enhanced engagement in physical activities, in turn, provides a fertile ground for the development of perseverance, emotional regulation, teamwork, and self-control, which are recognized as critical dimensions of non-cognitive abilities. Consequently, this study proposes H4: Social support and physical exercise play a chain mediating role between parent-child communication and adolescents' non-cognitive abilities.

Current research has thoroughly recognized the importance of parental involvement in the development of adolescents' non-cognitive abilities. However, three common limitations persist. First, as the most direct manifestation of parental involvement, parent-child communication is crucial for adolescent development. However, existing studies have primarily focused on the broader effects of parental involvement on non-cognitive abilities, with limited attention paid to the specific domain of parent-child communication. Second, most research relies on samples from single regions, limiting the representativeness of the conclusions. Even studies utilizing national datasets often lack in-depth exploration of regional differences and variations in the effects of parent-child communication across adolescents with diverse characteristics. Third, there is a scarcity of research examining the mediating mechanisms through which parent-child communication influences adolescents' non-cognitive abilities, leaving this relationship largely a "black box."

In light of these gaps, this study seeks to address these limitations by verifying the causal effects and underlying mechanisms of parent-child communication on adolescents' non-cognitive abilities and examining the heterogeneity of these effects across different adolescent characteristics. Accordingly, the hypothesis model is illustrated in Fig 1.

## Method

### Data sources

This study analyzes data from the China Education Panel Survey (CEPS), designed and implemented by the China Survey and Data Center at Renmin University of China. The CEPS dataset employs a combination of stratified, multi-stage, probability-proportional-to-size (PPS) sampling techniques to conduct questionnaire surveys among middle school students and their parents, teachers, and school administrators. The 2013–2014 baseline survey covered 28 county-level administrative units, with 112 schools and 438 classes, and successfully collected data from 19,487 seventh-grade students. In 2014–2015, the CEPS data underwent a follow-up survey, successfully re-surveying 9,449 original samples while adding 471 new eighth-grade student samples, resulting in a total of 9,920 samples for the 2014–2015 survey.

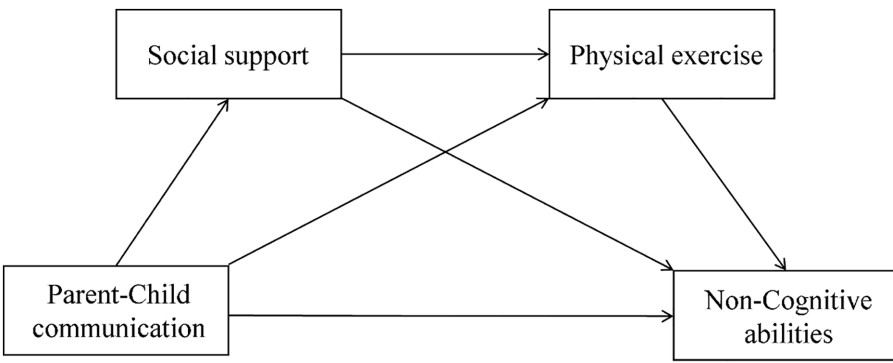

**Fig 1. Hypothesis model.**

Given the research theme and the need for timely data, this study specifically utilizes the 2014–2015 survey data, which included a follow-up of the seventh-grade students from the baseline survey. The dataset used in this study consists of eighth-grade student samples for this academic year, comprising adolescents approximately 14 years old. The questionnaire encompasses a wide range of topics, including respondents' basic demographic information, physical and mental health, school and community behavior, and family education practices. To meet the research requirements, samples with missing values were processed through exclusion and mean imputation, resulting in a final dataset of 5,055 observations. Research ethics approval for data collection in the China Education Panel Survey (CEPS) dataset was granted by the Institutional Review Board of Renmin University, PR China. The survey was conducted in accordance with the ethical guidelines laid down in the Declaration of Helsinki. All participants provided written informed consent before taking part in the survey.

## Measures

**Dependent variables.** There is currently no unified standard for measuring non-cognitive abilities in academic research. Commonly used measurement indicators in existing studies include the "Big Five Personality" model, self-esteem, and locus of control. Considering that the "Big Five Personality" model is widely recognized across disciplines as an international standard for measuring non-cognitive abilities [44], as well as its ability to encompass diverse and complex personality traits, this study adopts the model as its basis for measurement. Drawing on similar studies [10,45], this research utilizes the China Education Panel Survey (CEPS) questionnaire data and adopts the widely accepted five–factor model to construct proxy indicators for the following five categories of non-cognitive abilities:

(1) Conscientiousness. This dimension includes items such as: "Even if I feel slightly unwell, or there are other reasons to stay at home, I still try to go to school", "Even if it's a subject I don't like, I will do my best" and "Even if homework takes a long time to complete, I still keep trying my best."

(2) Agreeableness. This involves questions like: "In the past year, did you do the following: help elderly people with tasks, follow order and queue up consciously, and treat people sincerely and kindly?"

(3) Extraversion. This dimension includes items include: "I often sit alone rather than being with others," "When with classmates or peers, I don't talk much and mostly listen," and "I am usually confident about the tasks I need to complete."

(4) Openness. This dimension is assessed with questions such as: "I often participate in activities organized by the school or class," "I can persist with my hobbies and interests," and "There are adults I respect and admire."

(5) Neuroticism. The survey inquires whether students have felt the following in the past seven days: depressed, nervous, overly worried, unhappy, life being meaningless, sad, feeling that something bad is going to happen, inattentive in class. Likert scoring was used for all items.

The Cronbach's α for all items mentioned above is 0.779, indicating good reliability and validity. After the measurement items were determined, to eliminate differences in scale units among variables and facilitate the aggregation of subindicators, mean standardization was applied to the related items under the five personality dimensions on the basis of existing research and the database's information. This process yielded values for the five dimensions of non-cognitive abilities. The correlation analysis results (Table 2) indicate significant correlations among the five dimensions of non-cognitive abilities, suggesting that principal component analysis (PCA) can be used to synthesize an integrated non-cognitive ability index. Using PCA, two components with eigenvalues greater than 1 were extracted. These two components were then weighted and summed, with the weights corresponding to their respective variance contribution rates. The final integrated index of non-cognitive abilities was thus constructed.

**Independent variable.** Referring to similar studies [46], the frequency of parents actively discussing topics with their children, such as "what happened at school," "your relationships with friends," "your relationships with teachers," and "your concerns or worries," was averaged to calculate the parent-child communication score (1 = never, 2 = occasionally, 3 = often; Cronbach's α = 0.871).

## Control variables

Referring to existing studies [47], this paper incorporates factors at the individual adolescent, family and school levels as covariates of the study. (1) Individual level: Gender, whether an only child, and household registration (Hukou).(2) Family level: Family economic status, the highest educational level of parents (considering the higher education level between the two), and the number of books at home.(3)School level: school type (using the 2013–2014 School Administrator Questionnaire: A1. "What is the type of your school?" public school = 1, private school = 0), school quality (using the 2013–2014 School Administrator Questionnaire: A4. "In terms of educational performance, how does your school's junior high section currently rank within your county/district?" worst = 1, below average = 2, average = 3, above average = 4, best = 5), school location (using the 2013–2014 School Administrator Questionnaire: A23. " Where the school is located?" central urban areas of cities and counties = 1, central urban area = 1, peripheral urban or urban-rural area = 2, rural = 3. This item originally had 5 options in the questionnaire, but for simplified data, it was condensed into 3 categories according to the questionnaire manual.) and school facility resources. (using the 2013–2014 School Administrator Questionnaire: A12. "Whether the school had specific facilities, including a laboratory, computer room, library, music room, student activity room, psychological counseling room, cafeteria, sports field, gymnasium, and swimming pool". Each question was scored on a continuous scale from 1 to 3. The average score across the 10 items was calculated to create the school facility resources variable.)

## Mediation

For the "social support" variable, questions from the survey such as "Most of my classmates are friendly towards me," "I feel close to the people at this school," "I have adjusted from these emotions with the help of others," "When these emotions arise, teachers will try to help me," and "When these emotions arise, teachers will contact my parents to help me" are utilized. The term "these emotions" refers to the negative emotions mentioned in the Emotional Stability section. A 4-point Likert scale is used for these items, where 1 = strongly disagree, 2 = disagree, 3 = agree, and 4 = strongly agree, Cronbach's α = 0.891. The average score of these five questions is calculated, with higher values indicating greater perceived social support.

For the "physical exercise" variable, the total weekly exercise duration is calculated by multiplying the number of exercise days per week by the daily exercise duration. Extreme cases where the duration of a single session of physical

exercise exceeds 360 minutes (or 6 hours) are excluded. The total weekly exercise duration is then divided by 7 to determine the average daily exercise duration. Finally, a logarithmic transformation is applied to this average daily duration to smooth the data, resulting in the study's independent variable of exercise duration.

## Statistical methods

This study utilized Stata 17.0 and SPSS 26 for data analysis, presenting measurements as means and standard deviation. Initially, descriptive statistics were performed to summarize the basic information of the sample. Then Pearson correlation analysis were carried out. Finally, Bootstrap testing of the chain mediation effects of social support and physical exercise were undertaken using the PROCESS plug-in (Model 6) in SPSS. The sample size was set to 5000 iterations, and the confidence interval was set to 95%.

# Results

## Descriptive statistics

Descriptive statistics of each variable are shown in Table 1. The sample consists of 51.71% male students, and 55.37% of the respondents came from rural areas. Among all respondents, 40.73% are only children and the average parent-child communication level is 2.075. For school type, the vast majority of students attended public schools. The average family economic status is moderate, and the parental education level shows considerable variation.

## Bivariate correlations of the key variables

The Pearson correlation coefficient of the key variables in this study are shown in Table 2. The results of correlation analysis indicated that parent-child communication was significantly positively correlated with social support ($r=0.336$, $p<0.001$), physical exercise ($r=0.186$, $p<0.001$) and non-cognitive abilities ($r=0.301$, $p<0.001$); Likewise, social support was significantly positively correlated with both physical exercise ($r=0.176$, $p<0.001$) and non-cognitive abilities ($r=0.386$, $p<0.001$). Additionally, physical exercise was significantly positively correlated with non-cognitive abilities ($r=0.197$, $p<0.001$).

**Table 1. Descriptive statistics.**

| Variables | Definition | Mean/Percentage | SD |
|---|---|---|---|
| Physical Exercise | Logarithm of average daily physical exercise time | 1.074 | 0.703 |
| Parent-child communication | Average frequency of communication between parents and children | 2.075 | 0.517 |
| Non-cognitive Abilities | Composite index reflecting non-cognitive abilities | 0.000 | 0.734 |
| Social Support | Average of 5 items indicating social support | 2.593 | 0.439 |
| Gender | Male=1, Female=0. | 51.71% | — |
| Only child | Yes=1, No=0 | 40.73% | — |
| Household Registration | Rural=1, Urban=0 | 55.37% | — |
| Family Economic Status | Very difficult – Very wealthy 5-level rating | 2.924 | 0.605 |
| Parental Education Status | The highest education level achieved by either the mother or the father | 4.634 | 2.076 |
| Household Book Collection | Few – Many 5 ratings | 3.029 | 1.158 |
| School Type | Public school=1, Private school=0 | 92.68% | — |
| School Quality | Worst-Best 5-level rating | 3.880 | 0.809 |
| School Facility Resources | None=1, Available but needs improvement=2, Available and good=3 | 2.025 | 0.559 |
| School Location | Central urban areas of cities and counties=1, Central urban area=1, Peripheral urban or urban-rural area=2, Rural=3 | 2.002 | 0.862 |

**Table 2. Correlation analysis results of key variables(n = 5055).**

| Variables | Parent-child communication | Social support | Physical exercise | Non-cognitive abilities |
|---|---|---|---|---|
| Parent-child communication | 1 | – | – | – |
| Social support | 0.336*** | 1 | | – |
| Physical exercise | 0.186*** | 0.176*** | 1 | – |
| Non-cognitive abilities | 0.301*** | 0.386*** | 0.197*** | 1 |

Note. *p < 0.01. **p < 0.05.***p < 0.001.

## Heterogeneity analysis

We have verified the positive impact of parent-child communication on adolescents' non-cognitive abilities and further explored whether this effect varies across individual characteristics, focusing on household registration status and only child status.

The heterogeneity analysis results (Table 3) show that parent-child communication significantly influences non-cognitive abilities of both rural ($r = 0.313$, $P < 0.001$) and urban ($r = 0.435$, $P < 0.001$) adolescents, with a stronger effect observed in urban areas. This disparity may be attributed to the dual urban–rural societal structure. Urban adolescents are more likely to live with their parents, who tend to have higher educational levels, enabling more frequent and higher–quality parent-child interactions. In contrast, rural parents often lack awareness of the importance of communication and adhere to traditional parenting practices, which may hinder close emotional connections [48]. Additionally, rural adolescents frequently face challenges such as being "left behind" due to parental migration for work, significantly reducing opportunities for meaningful parent-child communication and thereby weakening its role in developing non-cognitive abilities.

Similarly, the effect of parent-child communication differs by only–child status. Compared to non–only children ($r = 0.351$, $P < 0.001$), only children experienced a stronger positive influence ($r = 0.388$, $P < 0.001$). This finding aligns with prior research, which suggests that only–child families allocate more resources, time, and energy to their children. Parents in such families tend to spend more time accompanying their children, fostering closer relationships and more frequent communication [49]. Furthermore, under the constraints of limited family resources, only children receive a larger share of parental attention and care, making them more likely than their nonchild peers to benefit from parent-child communication in the development of noncognitive abilities.

**Table 3. Heterogeneity analysis results.**

| | (1) Rural | (2) Urban | (3) Only child | (4) Not only child |
|---|---|---|---|---|
| Parent-child communication | 0.313*** | 0.435*** | 0.388*** | 0.351*** |
| | (0.028) | (0.031) | (0.032) | (0.028) |
| Control variables | Yes | Yes | Yes | Yes |
| _cons | −1.024*** | −1.380*** | −1.480*** | −1.003*** |
| | (0.130) | (0.152) | (0.166) | (0.131) |
| N | 2799 | 2256 | 2059 | 2996 |
| $R^2$ | 0.091 | 0.138 | 0.135 | 0.089 |

Note. * p < 0.01. ** p < 0.05.*** p < 0.001.Robust standard error in parentheses.

## Mediation effect analysis

After controlling for individual–level, family–level and school–level variables, a mediation effect test procedure was employed to investigate the mediating role of social support and physical exercise in the connection between parent-child communication and adolescents' non-cognitive abilities. The results, as presented in Table 4 and Fig 2, indicate that parent-child communication has a significant positive predictive effect on adolescents' non-cognitive abilities ($r = 0.369$, $p < 0.001$). After incorporating social support and physical exercise as mediating variables, the positive predictive effect of parent-child communication on adolescents' non-cognitive abilities remains significant ($r = 0.215$, $p < 0.001$). Furthermore, parent-child communication significantly promotes adolescents' acquisition of social support ($r = 0.265$, $p < 0.001$) and increased their physical exercise time ($r = 0.154$, $p < 0.001$). Adolescents' social support significantly and positively predicts their physical exercise time ($r = 0.192$, $p < 0.001$) and non-cognitive abilities ($r = 0.503$, $p < 0.001$). Similarly, adolescents' physical exercise also significantly and positively predicts their non-cognitive abilities ($r = 0.098$, $p < 0.001$).

The mediation analysis results (Table 5) indicate that parent-child communication had a significant direct effect on adolescents' non-cognitive abilities, with an effect size of 0.220. Social support and physical exercise mediate the relationship between parent-child communication and non-cognitive abilities through three pathways. (1) Social support serves as a mediator, with an indirect effect value of 0.131, accounting for 35.50% of the variance. (2) Physical exercise acts as a mediator, with an indirect effect value of 0.014, representing 3.79% of the variance. (3) Social support and physical

**Table 4. The chained mediation effect of social support and physical exercise.**

| Outcome Variables | Predictor Variable | β | SE | t | R² | F |
|---|---|---|---|---|---|---|
| Non-cognitive Abilities | Parent-child communication | 0.369 | 0.021 | 17.614*** | 0.114 | 54.879 |
| Social Support | Parent-child communication | 0.265 | 0.012 | 21.242*** | 0.120 | 54.580 |
| Physical Exercise | Parent-child communication | 0.154 | 0.022 | 6.867*** | 0.071 | 24.706 |
| | Social support | 0.192 | 0.030 | 6.471*** | — | — |
| Non-cognitive Abilities | Parent-child communication | 0.215 | 0.021 | 10.125*** | 0.207 | 90.830 |
| | Social support | 0.503 | 0.025 | 19.847*** | — | — |
| | Physical exercise | 0.098 | 0.109 | 6.816*** | — | — |

Note. *p < 0.01. **p < 0.05.***p < 0.001.

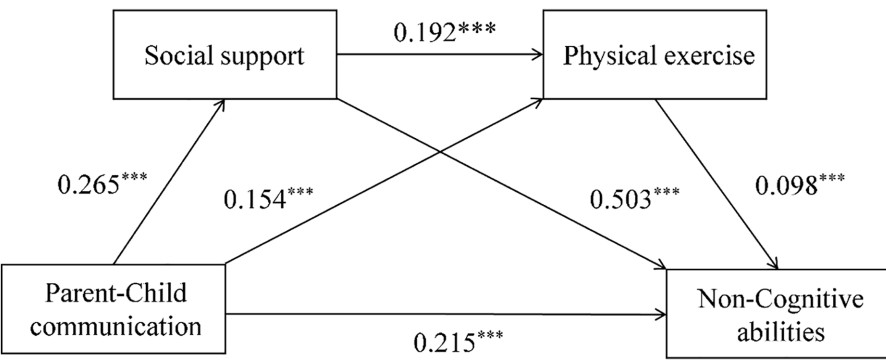

**Fig 2. The chained mediation model.**

**Table 5. Bootstrap analysis of the chain mediating effect test.**

| Effect | Pathways | Effect size | 95% CI | | SE | Ratio of effect |
|---|---|---|---|---|---|---|
| | | | Lower Limit | Upper Limit | | |
| Direct effect | parent-child communication→ non-cognitive abilities | 0.220 | 0.181 | 0.259 | 0.020 | 59.62% |
| Indirect effect | parent-child communication→ social support →non-cognitive abilities | 0.131 | 0.115 | 0.150 | 0.009 | 35.50% |
| | parent-child communication→ physical exercise →non-cognitive abilities | 0.014 | 0.009 | 0.020 | 0.003 | 3.79% |
| | parent-child communication→ social support→ physical exercise→ non-cognitive abilities | 0.004 | 0.003 | 0.006 | 0.001 | 1.09% |
| Total indirect effect | – | 0.149 | 0.132 | 0.169 | 0.009 | 40.38% |
| Total effect | – | 0.369 | 0.330 | 0.408 | 0.020 | 100% |

exercise operate as a chain mediator, yielding an indirect effect value of 0.004, which constitutes 1.09% of the variance. Notably, the 95% confidence intervals for all three mediation effects exclude zero, indicating their significance.

## Discussion

### The relationship between parent-child communication and adolescents' non-cognitive abilities

This study found that parent-child communication significantly had a positive impact on adolescents' non-cognitive abilities (Hypothesis 1), which is consistent with existing research findings, showing that close parent-child relationships enhance adolescents' self–efficacy and agreeableness [50,51]. The positive influence of parent-child communication can be attributed to several key aspects: firstly, it creates emotional security that fosters confidence and resilience in adolescents' daily challenges [52]; Second, it provides rich opportunities for social learning, enabling adolescents to observe and internalize positive behaviors and values from their parents [30]; Third, it creates a supportive environment that facilitates social–emotional skill development through continuous dialogue and interaction [19]. Notably, these effects were more pronounced among urban adolescents and only children, highlighting how family context and resources can shape the effectiveness of parent-child communication. During adolescence, a critical period for non-cognitive development, youth experiencing active parental engagement often demonstrate enhanced non-cognitive capabilities compared to those with limited parental interaction [53]. These findings remained robust after controlling for individual, family, and school–level variables, suggesting that parent-child communication serves as a crucial foundation for adolescent non-cognitive development. In practice, parents should prioritize maintaining open and supportive communication with their children, while schools could implement targeted programs to enhance parent-child interactions across different family contexts.

### The mediating role of social support

This study validated that social support partially mediates the relationship between parent-child communication and adolescents' non-cognitive abilities (Hypothesis 2). This finding aligns with the perspective of attachment theory, which posits that the "internal working model" formed through early parent-child relationships not only influences individuals' emotional experiences but also shapes their social interaction patterns, thereby fostering the development of non-cognitive abilities. Specifically, effective parent-child communication not only directly enhances adolescents' non-cognitive abilities but also indirectly exerts its influence by strengthening their perception and utilization of social support (the empirical analysis showed that the indirect effect through social support was 0.131, 95% CI = [0.115, 0.150], accounting for 35.50% of the total effect). This mediating role can be understood through two key pathways: firstly, parent-child communication creates a harmonious family atmosphere, enabling adolescents to acquire social skills such as trust and empathy in close relationships, which enhances their ability to access social support [54]. This supportive family environment encourages

adolescents to actively engage in social interactions and develop stronger non-cognitive abilities [55]. Second, for middle school students, support from teachers and peers holds unique significance. Positive teacher–student interactions help enhance adolescents' sense of responsibility and self–management skills[57], while strong peer relationships significantly improve self–esteem and cooperation[58], collectively promoting the development of non-cognitive abilities. Consequently, parent-child communication indirectly promotes adolescents' non-cognitive abilities by increasing the social support they receive.

## The mediating role of physical exercise

This study also confirmed the mediating role of physical exercise between parent-child communication and adolescents' non-cognitive abilities(Hypothesis 3). Mechanistically, effective parent-child communication not only enhances adolescents' awareness of physical exercise, encouraging them to place greater emphasis on physical exercise [31], but also strengthens family cohesion through interaction, providing a stable and harmonious family environment. In such an environment, adolescents are more likely to develop healthy lifestyle habits and actively engage in physical exercise [56]. Additionally, parental praise and encouragement can boost adolescents' self–confidence and self–efficacy, further increasing their willingness to participate in physical exercise [57]. Regular physical exercise is not only an essential component of a healthy lifestyle but also significantly promotes the development of adolescents' non-cognitive abilities. Physical exercise has been shown to improve adolescents' non-cognitive skills, with those possessing weaker non-cognitive abilities benefiting the most [14]. Moreover, consistent physical exercise over a certain period can enhance students' attention and resistance to distraction [58]. Therefore, parent-child communication promotes adolescents' non-cognitive abilities by encouraging physical exercise behaviors. However, in recent years, policies at various government levels have increasingly recognized the role of physical exercise in promoting adolescents' holistic development. These efforts aim to reduce the academic workload and extracurricular tutoring burdens of students in compulsory education [59]. Incorporating physical exercise into the model underscores its critical role in enhancing adolescents' non-cognitive abilities, providing policy-makers and educators with empirical evidence and data to inform decision–making and educational practices.

## Chain mediation of social support and exercise

This study reveals that social support and physical exercise serve as a chain mediation mechanism between parent-child communication and adolescents' non-cognitive abilities (Hypothesis 4). This chain relationship can be elucidated through the lens of Self–Determination Theory (SDT). Specifically, social support from parents and peers fulfills three fundamental psychological needs: autonomy, competence and relatedness. The satisfaction of these needs creates intrinsic motivation that transforms exercise intentions into sustained behavioral engagement, which aligns with the findings of Sallis et al [60]. According to the results of this study, in the chain mediation pathway between parent-child communication and adolescents' non-cognitive abilities, physical exercise mediates the relationship between social support and non-cognitive abilities. Contrary to the traditional beliefs held by many Chinese parents, this study reveals that physical exercise does not hinder academic performance. Instead, it has a positive effect on adolescents' physical and mental health, both of which are critical components of non-cognitive abilities [61]. Moreover, related studies confirm that physical exercise enhances adolescents' non-cognitive abilities, which in turn improves their cognitive abilities and ultimately promotes academic performance [62]. Given the role of social support in encouraging physical exercise and the positive effects of physical exercise on adolescents' non-cognitive abilities, this study concludes that social support can indirectly influence adolescents' non-cognitive abilities through physical exercise.

In this study, both social support and physical exercise significantly influence adolescents' non-cognitive abilities, yet their chain mediation effect appears relatively small. One possible explanation is that the strength of the chain mediation effect depends on the linkages between variables at each stage, and in the Chinese context, several structural and cultural factors may weaken these associations. First, under the highly competitive education system, adolescents often face

substantial academic pressure, leading them to devote the majority of their discretionary time to study rather than physical activity. This tendency restricts the extent to which social support can be translated into increased exercise participation [63].

Second, families in China frequently prioritize academic achievement as the primary channel of upward social mobility, thereby allocating fewer resources and less encouragement to sports and leisure activities. This prioritization further attenuates the effectiveness of social support in promoting physical exercise [64]. Third, opportunities for structured physical activity remain constrained by school curricula and the limited availability of extracurricular sports programs, which diminishes the potential impact of parental support on exercise behaviors [65]. Moreover, the influence of parental involvement on adolescents' physical activity may exhibit a temporal lag, with its effects often emerging only after several years [66].

Although the chain mediation effect accounts for a relatively small proportion, this finding underscores the complex interplay between psychological support and behavioral engagement. Importantly, it suggests that improving adolescents' non-cognitive abilities requires a more balanced educational approach. At the school level, policies should aim to reduce excessive academic burden and integrate physical education more effectively into the curriculum. At the family and community levels, efforts should be made to provide adolescents with accessible opportunities for exercise and to encourage a culture that values both academic success and physical well-being. Such initiatives would not only strengthen the mediating role of physical exercise but also create a more supportive environment for the holistic development of adolescents' non-cognitive abilities.

## Implications

This study revealed that parent-child communication has a significant impact on adolescents' non-cognitive abilities. By constructing a chain mediation model, this study explored the mediating roles of social support and physical exercise, further clarifying the relationships among parent-child communication, social support, physical exercise, and non-cognitive abilities. Parent-child communication creates a supportive environment for adolescents by fostering emotional connections and providing guidance, strengthening their social support networks. These enhanced networks motivate adolescents to actively participate in physical exercise, thereby improving traits such as extraversion and emotional stability, which are key components of non-cognitive abilities. These results have important implications for families, educators, and policymakers. Parents must engage in regular, positive interactions with their children, emphasizing active listening and emotional support. Adolescents need guidance to initiate communication with their parents, sharing thoughts and seeking advice on various life challenges. Educational institutions play a crucial role in fostering adolescents' non-cognitive abilities. Schools should implement programs that connect families and schools through regular parent-teacher communication. Integrating social-emotional learning into the curriculum is essential, as is training teachers to address students' emotional needs. Schools must also promote physical activity by maintaining robust physical education programs and offering diverse extracurricular sports. These strategies enable schools to create a comprehensive environment that promotes students' social support, physical exercise, and the development of non-cognitive abilities. Given the diverse backgrounds of adolescents, implementing targeted intervention strategies is crucial. These strategies must account for urban-rural disparities, family structures, and individual personality traits. The government should implement targeted policies for rural areas and multi-child families. These should include parental communication training in rural regions and additional educational and sports subsidies for families with multiple children. Specific measures could involve hosting family education lectures at rural community centers, establishing education funds for multi-child families, and developing public sports facilities in rural areas. These initiatives aim to enhance parenting skills and provide equal educational and physical activity opportunities for all children.

## Limitations

This study has certain limitations. First, owing to the constraints of the dataset, the sample is focused on adolescents within a specific age range, and the applicability of the findings to other age groups requires further validation.

Second, the path analysis in this study primarily examines the roles of social support and physical exercise. Future research should broaden its scope to explore additional pathways through which parent-child communication influences adolescents' non-cognitive abilities, as well as the boundary conditions under which these effects occur, to provide a more comprehensive perspective of related theories and practices. Third, the study's reliance on self–reported data and the " Big Five Personality " model to assess adolescents' non-cognitive abilities may introduce measurement bias. Future research should employ more robust scales and incorporate teacher and parental evaluations to enhance measurement accuracy. Lastly, this study relies on data from the 2014–2015 survey, and as a cross–sectional study, its causal inferences require further support and validation through experimental or longitudinal research.

## Conclusion

While numerous studies have highlighted the beneficial impacts of parent-child communication on adolescents' cognitive skills, academic outcomes, and mental health, the exploration into how such communication affects adolescents' non-cognitive abilities remains relatively rare. This paper leverages data from the 2014–2015 China Education Panel Survey (CEPS) to investigate how parent-child communication positively influences adolescents' non-cognitive abilities and the mechanisms behind this influence. The findings reveal that regular parent-child communication substantially boosts adolescents' non-cognitive abilities, with social support and physical exercise serving as independent and joint mediators in this relationship, forming a chain mediation effect. It is recommended that parents strengthen communication with their adolescent children through various methods, focusing on effective communication styles and techniques to enhance the quality of parent-child interactions. Additionally, efforts should be made to improve the level of social support available to adolescents and to encourage their active participation in physical exercise, thereby fostering the development of non-cognitive abilities. These measures can help middle school students navigate adolescence smoothly, promoting their physical and mental well–being as well as holistic development.

## Supporting informtion

**S1 File. Original Data and Cleaned Data.**
(ZIP)

## Author contributions

**Conceptualization:** Huijie Guo.

**Data curation:** Minghan Cai.

**Funding acquisition:** Huijie Guo, Zhiqiang Zheng.

**Investigation:** Zhengmei Lin.

**Methodology:** Minghan Cai.

**Resources:** Dailong Fang, Zhiqiang Zheng.

**Software:** Weifeng Guo.

**Supervision:** Dailong Fang, Yao Zheng.

**Validation:** Weifeng Guo.

**Visualization:** Zhengmei Lin.

**Writing – original draft:** Minghan Cai.

**Writing – review & editing:** Huijie Guo.

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
