## [Decision Letter · Decision Letter 0]

11 Sep 2025

Dear Dr. Guo,

Thank you for submitting your manuscript to PLOS ONE. After careful consideration, we feel that it has merit but does not fully meet PLOS ONE’s publication criteria as it currently stands. Therefore, we invite you to submit a revised version of the manuscript that addresses the points raised during the review process.

We look forward to receiving your revised manuscript.

Kind regards,

Henri Tilga, PhD

Academic Editor

PLOS ONE

**Journal Requirements:**

1. When submitting your revision, we need you to address these additional requirements. Please ensure that your manuscript meets PLOS ONE's style requirements, including those for file naming. The PLOS ONE style templates can be found at https://journals.plos.org/plosone/s/file?id=wjVg/PLOSOne_formatting_sample_main_body.pdf and https://journals.plos.org/plosone/s/file?id=ba62/PLOSOne_formatting_sample_title_authors_affiliations.pdf 2. Thank you for stating in your Funding Statement: This work was supported by the National Office for Philosophy and Social Science of China [grant number 22BTY114].  Please provide an amended statement that declares *all* the funding or sources of support (whether external or internal to your organization) received during this study, as detailed online in our guide for authors at http://journals.plos.org/plosone/s/submit-now.  Please also include the statement “There was no additional external funding received for this study.” in your updated Funding Statement. Please include your amended Funding Statement within your cover letter. We will change the online submission form on your behalf. 3. Please amend the manuscript submission data (via Edit Submission) to include author Dr. Yao Zheng. 4. Your ethics statement should only appear in the Methods section of your manuscript. If your ethics statement is written in any section besides the Methods, please move it to the Methods section and delete it from any other section. Please ensure that your ethics statement is included in your manuscript, as the ethics statement entered into the online submission form will not be published alongside your manuscript. 5. Please remove your figures from within your manuscript file, leaving only the individual TIFF/EPS image files, uploaded separately. These will be automatically included in the reviewers’ PDF. 6. If the reviewer comments include a recommendation to cite specific previously published works, please review and evaluate these publications to determine whether they are relevant and should be cited. There is no requirement to cite these works unless the editor has indicated otherwise. 

Reviewers' comments:

**Comments to the Author**

1. Is the manuscript technically sound, and do the data support the conclusions?

Reviewer #1: Yes

2. Has the statistical analysis been performed appropriately and rigorously?

Reviewer #1: No

3. Have the authors made all data underlying the findings in their manuscript fully available?

Reviewer #1: Yes

4. Is the manuscript presented in an intelligible fashion and written in standard English?

Reviewer #1: No

**Reviewer #1: ** This manuscript explores the relationship between parent-child communication and adolescents’ non-cognitive abilities, with a focus on the mediating roles of social support and physical exercise.

1.The rationale for the chain mediation hypothesis (social support → physical exercise → non-cognitive abilities) is underdeveloped. While prior studies are cited, the theoretical mechanism is not sufficiently clear. Please expand the theoretical discussion, perhaps drawing on Self-Determination Theory or Ecological Systems Theory, to clarify why social support should lead to more physical exercise, which in turn enhances non-cognitive abilities.

2.The results show that the chain mediating effect of physical exercise is relatively small. This is an important finding, but it is currently under-explained. It is suggested to conduct an in-depth analysis of the reasons, such as the high academic pressure on students under the Chinese education system, the limitation of exercise time, or the priority given to academic investment by families, and further propose educational and policy implications."

3.The abstract is overly detailed, reporting multiple coefficients and percentages. Consider simplifying by emphasizing the main results and contributions, while moving detailed statistics into the Results section.

4.The review of social support and physical exercise includes overlapping points and lengthy descriptions. Streamlining these sections would improve readability and sharpen the theoretical contribution. Some references are incomplete or inconsistently formatted. Please carefully check against journal guidelines.

**Do you want your identity to be public for this peer review?** For information about this choice, including consent withdrawal, please see our Privacy Policy

Reviewer #1: No

---

## [Author Response · Author response to Decision Letter 1]

28 Oct 2025

Reviewer#1

Comment 1:

The rationale for the chain mediation hypothesis (social support → physical exercise → non-cognitive abilities) is underdeveloped. While prior studies are cited, the theoretical mechanism is not sufficiently clear. Please expand the theoretical discussion, perhaps drawing on Self-Determination Theory or Ecological Systems Theory, to clarify why social support should lead to more physical exercise, which in turn enhances non-cognitive abilities.

Response 1:

Line 127-134,page 6

We sincerely thank the reviewer for this valuable comment. In the revised manuscript, we have expanded the theoretical discussion of the chain mediation hypothesis by incorporating Self-Determination Theory (SDT) to clarify the underlying mechanism linking social support, physical exercise, and non-cognitive abilities. Specifically, we now argue that social support satisfies adolescents’ basic psychological needs for relatedness, competence, and autonomy, which fosters the internalization of positive attitudes toward physical activity and strengthens intrinsic motivation for participation. In turn, sustained engagement in physical exercise provides opportunities for the development of perseverance, emotional regulation, teamwork, and self-control, key dimensions of non-cognitive abilities. These theoretical additions help clarify why social support promotes physical exercise and how this process ultimately contributes to adolescents’ non-cognitive development, thus addressing the reviewer’s concern regarding the theoretical foundation of the mediation pathway.

Comment 2:

The results show that the chain mediating effect of physical exercise is relatively small. This is an important finding, but it is currently under-explained. It is suggested to conduct an in-depth analysis of the reasons, such as the high academic pressure on students under the Chinese education system, the limitation of exercise time, or the priority given to academic investment by families, and further propose educational and policy implications.

Response 2:

Line 376-396, page 20

We sincerely appreciate the reviewer’s insightful comment. In the revised manuscript, we have substantially expanded the discussion to provide a more in-depth interpretation of why the chain mediating effect of physical exercise appears relatively small. Specifically, we have added an analysis of several contextual factors in China’s educational and family systems that may weaken the linkage between social support and physical exercise. These include the high academic pressure under a competitive education system, the limited discretionary time for physical activity, and families’ prioritization of academic investment as a primary means of upward mobility. We have also discussed the limited availability of structured extracurricular sports programs and the potential temporal lag in parental influence on adolescents’ exercise participation. Furthermore, we have incorporated corresponding educational and policy implications, emphasizing the need to balance academic and physical development through school-level curriculum reforms and family- and community-based initiatives. These revisions collectively provide a more comprehensive and theoretically grounded explanation for the small chain mediation effect and enhance the practical relevance of our findings.

Comment 3:

The abstract is overly detailed, reporting multiple coefficients and percentages. Consider simplifying by emphasizing the main results and contributions, while moving detailed statistics into the Results section.

Response 3:

Line 23-30 page 1

Thank you for your valuable suggestion. Following your advice, we have revised the abstract by removing excessive statistical details such as coefficients and effect sizes, which are now presented in the Results section. The abstract has been streamlined to emphasize the main findings and overall contributions of the study, making it more concise and reader-friendly.

Comment 4:

The review of social support and physical exercise includes overlapping points and lengthy descriptions. Streamlining these sections would improve readability and sharpen the theoretical contribution. Some references are incomplete or inconsistently formatted. Please carefully check against journal guidelines.

Response 4:

Line 84-92+102-116, page 4-5

We sincerely thank the reviewer for this valuable suggestion. In the revised manuscript, we have streamlined and restructured the sections on social support and physical exercise to enhance coherence and readability. Specifically, we removed overlapping content, reduced redundant descriptions, and reorganized the theoretical logic to better align with the hypothesized mediation model. The discussion of social support now focuses on its conceptual definition, sources (family, teachers, peers), and psychological mechanisms through which it influences adolescents’ non-cognitive abilities, with a concise link to parent-child communication. The section on physical exercise has been refined to highlight two primary transmission channels—cognitive transmission and behavioral synchronization—through which parent-child communication affects adolescents’ exercise participation, followed by a clearer articulation of how exercise contributes to non-cognitive development from a bio-psycho-social and ecological systems perspective.

Additionally, all references have been carefully reviewed and reformatted in strict accordance with the journal’s citation style to ensure consistency and completeness. These revisions significantly improve the manuscript’s theoretical clarity, logical flow, and overall readability.

---

## [Decision Letter · Decision Letter 1]

30 Nov 2025

The association between parent-child communication and adolescents’ non-cognitive abilities: an examination of the chain mediating effect

PONE-D-25-40105R1

Dear Dr. Zheng,

We’re pleased to inform you that your manuscript has been judged scientifically suitable for publication and will be formally accepted for publication once it meets all outstanding technical requirements.

Kind regards,

Henri Tilga, PhD

Academic Editor

PLOS ONE

Additional Editor Comments (optional):

Reviewers' comments:

Reviewer's Responses to Questions

**Comments to the Author**

Reviewer #1: All comments have been addressed

2. Is the manuscript technically sound, and do the data support the conclusions?

Reviewer #1: Yes

3. Has the statistical analysis been performed appropriately and rigorously?

Reviewer #1: Yes

4. Have the authors made all data underlying the findings in their manuscript fully available?

Reviewer #1: Yes

5. Is the manuscript presented in an intelligible fashion and written in standard English?

Reviewer #1: Yes

Reviewer #1: Although I still have doubts about the sampling in this study, it does not affect my decision. I recommend accepting this paper.

**Do you want your identity to be public for this peer review?** For information about this choice, including consent withdrawal, please see our Privacy Policy

Reviewer #1: No

---

## [Editor Report · Acceptance letter]

PONE-D-25-40105R1

PLOS One

Dear Dr. Zheng,

I'm pleased to inform you that your manuscript has been deemed suitable for publication in PLOS One. Congratulations! Your manuscript is now being handed over to our production team.

Kind regards,

on behalf of

Dr. Henri Tilga

Academic Editor

PLOS One